# Psychological and Physical Changes Caused by COVID-19 Pandemic in Elementary and Junior High School Teachers: A Cross-Sectional Study

**DOI:** 10.3390/ijerph19137568

**Published:** 2022-06-21

**Authors:** Nobuyuki Wakui, Nanae Noguchi, Kotoha Ichikawa, Chikako Togawa, Raini Matsuoka, Yukiko Yoshizawa, Shunsuke Shirozu, Kenichi Suzuki, Mizue Ozawa, Takahiro Yanagiya, Mayumi Kikuchi

**Affiliations:** 1Division of Applied Pharmaceutical Education and Research, Faculty of Pharmaceutical Sciences, Hoshi University, 2-4-41 Ebara, Shinagawa-ku, Tokyo 142-8501, Japan; s181180@hoshi.ac.jp (N.N.); s181012@hoshi.ac.jp (K.I.); s181156@hoshi.ac.jp (C.T.); s181219@hoshi.ac.jp (R.M.); y-yoshizawa@satsuki-ph.com (Y.Y.); s-shirozu@hoshi.ac.jp (S.S.); kenichi-suzuki@hoshi.ac.jp (K.S.); 2Shinagawa Pharmaceutical Association, 2-4-2 Nakanobu, Shinagawa-ku, Tokyo 142-0053, Japan; mitziozza1965@gmail.com (M.O.); yanagiya@tnb.co.jp (T.Y.); tomato_5mk@yahoo.co.jp (M.K.)

**Keywords:** COVID-19, elementary teacher, junior high school teacher, psychological change, physical changes

## Abstract

This study aimed to determine psychological and physical differences in elementary and junior high school teachers during COVID-19. This questionnaire-based cross-sectional study was conducted among 427 teachers in Tokyo, Japan (between 15 and 30 October 2020). The questionnaire explored school type (elementary and middle schools), sex, age, and COVID-19 changes (psychological changes, physical changes, impact on work, and infection control issues perceived to be stressed). Post hoc tests for *I cannot concentrate on work at all*, found a significant difference for *no change–improved* and *male teacher in elementary school female teacher in junior high school* (*p* = 0.03). Regarding *stress situation due to implementation of COVID-19 infection control*, there were significant differences for *disinfection work by teachers* between *male teachers in elementary school female teachers in junior high school* (*p* = 0.04) and *female teachers in elementary school female teachers in junior high school* (*p* = 0.03). COVID-19 produced differences in psychological and physical changes between male and female teachers in elementary and junior high schools. Some experienced psychological and physical stress, whereas others showed improvement. Given that teachers’ mental health also affects students’ educational quality, it is important to understand and improve teachers’ psychological and physical circumstances and stress.

## 1. Introduction

A new type of coronavirus occurred in Wuhan City, Hubei Province, China, in December 2019, [1] which was later named SARS-CoV-2 [2,3]. SARS-CoV-2 rapidly spread worldwide, and the World Health Organization (WHO) declared a pandemic in March 2020 due to the spread of COVID-19 caused by SARS-CoV-2 [4]. Subsequent lockdown restrictions occurred throughout the world, which introduced significant changes and confusion to daily life [5,6]. Currently, the total number of infected people worldwide is more than 560 million, and the number of deaths is more than 6 million [7]. 

In Japan, an emergency declaration was issued for the first time in April 2020 [8], and the government requested that restaurants and schools should close and working from home was encouraged [9,10]. A new way of life was proposed to reduce the spread of infection, and a comprehensive behavioral change was required [11]. Specifically, basic infection control measures, such as hand washing and physical distance [12,13], as well as routine infection control measures, such as thorough avoidance of three Cs (closed spaces, crowded places, close-contact settings) [14] and a reduction in human contact by 80%, were implemented [15]. Such measures were also introduced in many parts of the world [16,17], which is a stressor for many people [18]. 

A new method of education in schools was introduced to ensure the rights of children to receive an education [19]. Specifically, in addition to basic infection control measures, restrictions included eating silently and the promotion of online classes [19]. However, it has been recognized that these restrictions impose a physical and mental burden on teachers managing the learning environment of students [20], which affects the mental health of teachers [21]. 

In previous studies, education has been considered as one of the highly stressful occupations [22]; moreover, occupational stress affects mental and physical health, and increases the risk of developing conditions, such as depression [23]. Indeed, teachers have a higher risk of developing depression compared with other occupations [24], and the development of burnout syndrome due to accumulated stress has also been reported [25]. During the COVID-19 pandemic, teachers may be more highly stressed than previously [26] as they are required to take various measures to prevent the spread of infection in addition to their own personal safety [27]. 

In a survey conducted in Greece in 2020, changes were found in teachers’ mind and body related to COVID-19, and results showed that many teachers felt anxious due to the COVID-19 pandemic [28]. Compared with before the pandemic, mental changes were associated with decreased QOL [29], increased stress [30], and increased burnout [30,31]. There are some reports that female faculty are more likely to develop fear, depression, stress disorders and depression in comparison of sex [32]. In terms of physical changes, worsening of diet, sleep, and alcohol consumption have also been reported [6,17,18].

However, these reports have not analyzed the differences between elementary school teachers and junior high school teachers, and the stress situation of teachers by school type has not been clarified. Teachers are required to provide support for elementary school pupils and junior high school pupils in accordance with the students’ developmental stage; hence, the nature of their work and the way they interact with pupils should differ [33]. This has led to the possibility that the stress status of teachers involved in teaching during the COVID-19 pandemic can vary widely between elementary and junior high schools. Therefore, it is of crucial public health importance to clarify the stress situation of teachers by school type to improve teachers’ mental health [34]. 

Therefore, this study conducted a cross-sectional survey of elementary and junior high school teachers to clarify the physical and psychological differences during the COVID-19 pandemic.

## 2. Materials and Methods

### 2.1. Study Design and Participants

This was a cross-sectional study conducted among elementary and junior high school teachers in the Shinagawa Ward of Tokyo, Japan. The questionnaire (see Appendix A) was designed to be answered in about 10 min. The participants included elementary and junior high school teachers in Shinagawa Ward, Tokyo. Data were collected for over 16 days between 15 October 2020 and 30 October 2020. Questionnaires were distributed to teachers of 18 elementary schools and 11 junior high schools selected randomly, and collection was by mail. Teachers who participated in the study responded anonymously to the questionnaire. Study participants provided informed written consent and participants participated in the study voluntarily (Figure 1).

### 2.2. Survey Items

The questionnaire consisted of five sections, one for demographic data (age, sex, type of school), and four sections asked about the impact of COVID-19 (psychological changes, physical changes, impact on work, and stressed relating to infection control items).

Of the four sections of the questionnaire, the psychological change, physical change, and impact on work sections were prepared using items related to this study from the Disaster Reliever Mental Health Manual prepared in Japan [35]. The “infection countermeasure matter that causes stress” question item was made to be the matter for which the implementation is required in the infection countermeasure guideline for school education [36]. All the questions in the four sections investigating the impact of COVID-19 were closed questions.

In terms of psychological change, 13 items were investigated, including, feeling high, irritation, anger, indignation, regret, anxiety, chagrin, impotent feelings, depression, no sense of reality, no sense of time, emotions are paralyzed, unable to concentrate on work at all, and no longer wanting to engage with others.

Six physical changes were investigated: insomnia/nightmares, palpitations, standing dizziness, sweating, dyspnea, and digestive symptoms. The following four items were investigated regarding the effect on work: excessively immersed in the work, reduction in thinking ability, lower concentration ability, and lower work efficiency. These items were measured at three levels: more improved than before, same as before, more deteriorated than before.

As for the infection countermeasures that felt stress, nine items were surveyed, namely always wearing a mask, body temperature and health status, disinfection work by teachers, social distancing, activity restriction during class, using face shields, prohibition or restriction of play and conversation, eating without talking/no refills, and not allowing children to serve. These items were measured at two levels—feeling stressed and not feeling stressed. Subsequently, it was investigated whether there were significant differences between the types and sex of schools according to these questionnaire responses.

### 2.3. Sample Size

An online sample size calculator, Raosoft^®^ was used to set the sample size. Sample sizes were calculated using 50% response rate, 95% confidence interval, 5% error margin, and the population of teachers in elementary and middle schools in Tokyo as approximately 49,000 [37] (elementary school, *n* = 33,912, junior high school Rao soft, n = 15,340). As a result, the sample size recommended was 382. Therefore, the number of participants surveyed in this study was 427, which included an adequate sample size.

### 2.4. Statistical Analysis

Free statistical software R version 4.0.5 (R Core Team, R Foundation for Statistical Computing, Vienna, Austria) was used for statistical analysis. Pairwise elimination was used to exclude missing data from the analysis if any of the two variables had missing data, to use as much data as possible. Categorical data were tabulated with frequency and percentage, and continuous data were tabulated with mean ± SD. Differences in characteristics by school type and sex were assessed by Fisher’s exact test. In addition, for items with significant differences, multiple comparison tests of proportions were performed using the Hochberg methods. The level of significance in all tests was set at *p* < 0.05.

### 2.5. Ethical Considerations

The study was conducted with the approval of the Ethics Committee of the Hoshi University (approval no. 2020-05). The study was conducted in accordance with the Declaration of Helsinki Ethical Guidelines for Medical Research Involving Human Subjects. All participants provided written informed consent.

## 3. Results

### 3.1. Demographic Features

There were 427 participants, of whom 305 (71.4%) were elementary school teachers and 122 (28.6%) were junior high school teachers (Table 1). A total of 418 participants responded to sex—157 (37.6%) males and 261 (62.4%) females. A total of 403 participants responded to age—92 (22.8%) in their 20 s, 139 (34.5%) in their 30 s, 79 (19.6%) in their 40 s, 64 (15.9%) in their 50 s, and 29 (7.2%) in their 60 s.

### 3.2. Changes in Psychological Conditions before and after School Closure Due to COVID-19

Changes in psychological status before and after school closures are presented in Table 2. Fischer’s exact test revealed significant differences in the item “I cannot concentrate on work at all” (*p* = 0.02). In addition, when multiple comparison tests were performed as a post hoc test for I cannot concentrate on work at all, a significant difference was found in the part of no change–improved and male teacher in elementary school female teacher in junior high school (*p* = 0.03). Regarding the significant difference found in the item “I cannot concentrate on work at all,” junior high school teachers (males, 9.2%; females, 11.1%) had a higher response rate than elementary school teachers (males, 0%; females, 3.9%), indicating that they had improved. Furthermore, female teachers had a higher response rate to improvement than male teachers.

### 3.3. Changes in Physical Symptoms before and after School Closure Due to COVID-19

Changes in physical conditions before and after school closure are presented in Table 3. Fischer’s exact test showed significant differences for insomnia/nightmares (*p* = 0.009) and standing dizziness (*p* = 0.03). Regarding insomnia/nightmares, many female teachers of elementary (18.0%) and junior high schools (18.5%) answered that physical symptoms had deteriorated. Moreover, at the same time, a small number of female teachers responded that their insomnia/nightmare symptoms had improved (elementary female teacher: 7.8%, junior high school female teacher: 9.3%). Regarding standing dizziness, many female teachers in junior high school answered that it had deteriorated. In addition, a multiple comparisons test was performed for insomnia/nightmares and standing dizziness as a post hoc test, and no significant differences were found.

### 3.4. Impact of COVID-19 on Work before and after School Closure

The impact of COVID-19 on work before and after school closures presented in Table 4. Fischer’s exact test showed no significant differences among all four items. In sex comparisons, a greater proportion of female respondents stated that they had deteriorated in lowering of concentration ability compared with male teachers. In the comparison between the types of school, a large proportion of junior high school teachers answered that all four items deteriorated. In terms of lowering of work efficiency, >15% of the teachers indicated that they had deteriorated, especially junior high school teachers who indicated that they had deteriorated to a greater extent than elementary school teachers.

### 3.5. Stress Causes for Teachers Associated with COVID-19 Infection Control

The stress situation due to COVID-19 infection countermeasures is presented in Table 5. Fischer’s exact test showed significant differences in four items, namely, always wearing mask (*p* = 0.04), disinfection work by teachers (*p* = 0.03), activity restriction during class (*p* = 0.05), and prohibition or restriction of play and conversation (*p* = 0.04). In addition, when the multiple comparison tests were carried out as a post hoc test for the four items that showed significant differences, there were two significant differences in faculty disinfection work, between the two combination parts of male teachers in elementary school and female teachers in junior high school (*p* = 0.04) and female teachers in elementary school and female teachers in junior high school (*p* = 0.03).

## 4. Discussion

In Japan, the number of teachers suffering psychologically and physical increased more than twice over the 10-year period from 2002 to 2011, with an actual number of more than 5000 [38]. Worldwide, teachers’ occupational mental health problems are serious [24], and main teachers’ reasons for leaving are mental illness resulting from stress disorders and depression. Changes in schools due to COVID-19 have been linked with increased stress for faculty staff. This study assessed psychological and physical changes in teachers by school type and sex during the COVID-19 pandemic. The results showed significant differences in psychological symptom changes for *I cannot concentrate on work at all*, and in physical symptom changes for *insomnia/nightmares* and *standing dizziness*. Additionally, there were significant differences between males and females in elementary and junior high school teachers in the items of *always wearing a mask, disinfection work by teachers, activity restriction during class, and prohibition or restriction of play and conversation in stressful situations* resulting from the implementation of COVID-19 infection control measures. Furthermore, multiple comparison tests were performed on items for which a significant difference was found. As a result, two significant differences were found. The first was in the combination of elementary school male teachers–junior high school female teachers and improved–unchanged in *I cannot concentrate on work at all*. The second was in terms of stress—there was a significant difference between elementary school male teachers–junior high school female teachers and elementary school female teachers–junior high school female teachers in *disinfection work* by teachers.

The significant differences in the combination of female teachers in elementary school female teachers in male schools and improved–no change in *I cannot concentrate on work at all* of the psychological change may be attributed to the fact that the perceived stress and burden of elementary and middle school faculty was alleviated by COVID-19 spread. Previous investigations have reported that junior high school teachers are stressed by instructing students in club activities, and that elementary school teachers are stressed by the implementation of executive officials and committees [39]. Almost all students in junior high schools in Japan belong to clubs, and teachers provide guidance to students outside hours every day, including in holidays. In reality, junior high school teachers provide club activities even at the expense of holidays, and this is regarded as a cause of stress for teachers in Japan. With this in mind, it is possible that the burden on junior high school teachers was greatly reduced by the limitations on club activities due to the spread of COVID-19.

In Japan, more females than males work and are responsible for housework and parenting, and the decrease in teaching time for division activities requiring out-of-hours work is believed to directly affect the reduction in the burden on female teachers in junior high schools. Worldwide, it has also been reported that the burden of female teachers who are compatible with home and work is substantial [40,41], and the results of this survey may be transferrable to other countries.

In terms of physical changes, Fischer’s exact test showed significant differences in two items, *insomnia/nightmares* and *standing dizziness*. However, all post hoc tests for multiple comparisons showed no significant differences. In terms of *insomnia/nightmares*, elementary and junior high school female teachers showed higher improved response rates than male teachers. In contrast, there was a higher number of female teachers who responded that their *insomnia/nightmares* had deteriorated. Regarding this, the causes of *insomnia/nightmares* is believed to be stress, and females are more likely to feel psychological stresses, such as anxiety, than males. Therefore, while COVID-19 induces stress due to changes, at the same time, the reduction in workload may reduce stress. Regarding *standing dizziness*, the response rate of the participants who answered that the *standing dizziness* deteriorated was higher in middle school teachers than the elementary school teachers. For females, 0% of junior high school teachers and 7.8% of elementary school teachers responded that *standing dizziness* had improved, with the percentages of value differing by 7.8%. This may be because club activities are a major burden for junior high school teachers, and furthermore, female find balancing work with housework and childcare is stressful. However, some teachers answered that they improved in all items, and it is suggested that there some positive aspects to the spread of COVID-19.

No significant differences were found between school type and teacher sex in changes in *impact on work*. When checking each item, about 70% of teachers answered that there was no change in all items. Moreover, regardless of the sex and school type, approximately 10–20% of teachers responded that they had deteriorated in terms of *decreased thinking*, *concentration*, and *lowering of the work efficiency*, and a small percentage of teachers responded that they had improved. From this, it is suggested that the deterioration of thinking ability, concentration, and work efficiency occurred in some school teachers irrespective of the type of school and sex because anxiety [27] was newly generated by COVID-19.

The Fischer’s exact test found significant differences in the items always wearing a mask, disinfection work by teachers, activity restriction during class, and prohibition or restriction of play and conversation in the stress situation resulting from the implementation of COVID-19 infection countermeasure. Furthermore, the results of the multiple comparison test revealed significant differences in teacher disinfection work. In always wearing a mask, a higher proportion of females felt stressed compared with males. This may be because make-up adheres to the mask and make-up is spoiled [42]; in addition, hairstyles do not last well under the moisture from wearing the mask [43], and females report feeling stress from this.

Regarding disinfection work by teachers, a higher proportion of junior high school teachers reported feeling stressed compared with elementary school teachers. In addition, the percentage of females who answered that they were feeling stressed was higher than males. This is thought to be due in part to the fact that junior high schools have club activities and the range of activities is wider than that of elementary schools; therefore, the range of disinfection by teachers is also wider. In addition, generally, more females are more sensitive than males about their appearance and hands, consequently, the continuous use of disinfectants, which can cause rough hands [44], may make females more sensitive and susceptible to stress than in males. Moreover, given that females have lower physical strength than young males, they are more likely to feel the burden considering that the disinfection range is broad, including desks, chairs, rockers, and other items [36], which takes time and effort because carrying equipment, such as cleaning tools, can be a heavy burden.

Regarding the items of activity restriction during class and prohibition or restriction of play and conversation, a higher proportion of elementary school teachers felt stressed compared with junior high school teachers. Additionally, the percentage of males feeling stressed was higher than that of females. This is thought to have led to the burden and stress among elementary school teachers because there are more children in elementary school who are mentally younger than those in junior high school, and it is difficult to take command. Male elementary school teachers can be more stressed than female teachers because male teachers are more in charge of instructing students. Male teachers, in particular, often yell to discipline their students, which can be a reason for stress [45]. Equally, a number of teachers felt they had improved in some items, and it can be said that the spread of COVID-19 is not only bad but also that it has had a positive effect in some areas.

There are several limitations to this study. First, Fisher’s exact test showed significant differences, whereas the multiple comparison test showed that some of them did not significantly differ. As 18 tests were performed for a single item in this study’s multiple testing, it is possible that a so-called type 2 error occurred. Second, this study is aimed at teachers in Tokyo, and the opinions of teachers in other areas are not reflected. Thirdly, this study did not perform model analysis. Accordingly, the current study tested and reported the distribution of the results obtained during the COVID-19 pandemic and epidemic and the differences between them. Some of the dependent variables included herein had three levels, so all analyses were performed in a unified manner using Fisher’s exact test for ease of interpretation. Given that this survey found significant differences between the levels, further research considering the use of models, such as logistic regression analysis, can be considered in order to evaluate the factors. Nonetheless, this study is the first to assess differences in elementary and junior high schools among teachers and psychological and physical differences between sex due to the COVID-19 pandemic.

The emphasized point of our study is that previous reports have only assessed the relationship between sexes [29] or school differences [46] and psychological and physical effects, and no survey like this one has previously been conducted. In addition, it is interesting to evaluate not only the rate of increasing stress due to the spread of COVID-19 but also the rate of improvement. In fact, statistically significant differences between male and female teachers in elementary and junior high schools were also observed in the rate of improvement in this study. Reports evaluating the percentage of points that were improved by COVID-19 spread are unusual and remarkable. Even after COVID-19 has ended, it will be important to conduct surveys to assess the differences in improvements and deterioration between schools and sex, as well as overall improvement and deterioration, and to evaluate the causes objectively.

Teachers play an important role not only in their studies but also in the development of children’s mental health, as experts in education in daily contact with students and children [47]. Teachers’ mental health also affects students’ educational quality, so it is important to understand and improve teachers’ psychological and physical situations and stress. While the spread of COVID-19 has increased the stressors of many teachers, it is also possible that the work has been streamlined. It is recommended that administrative personnel involved in education and members of local educational committees conduct surveys such as this on a regular basis and assess the results to understand the mental health of faculty members and to strive to improve the working environment.

## 5. Conclusions

COVID-19 resulted in differences in psychological and physical changes between male and female teachers in elementary and junior high schools. While most teachers showed no change, some experienced psychological and physical stress, whereas others experienced improvement. Policymakers of administrative and educational committees should conduct surveys like this on teachers on a regular basis to understand the psychological and physical stress situations of teachers in a multidimensional manner as well as to decide the policy for the improvement.

## Figures and Tables

**Figure 1 ijerph-19-07568-f001:**
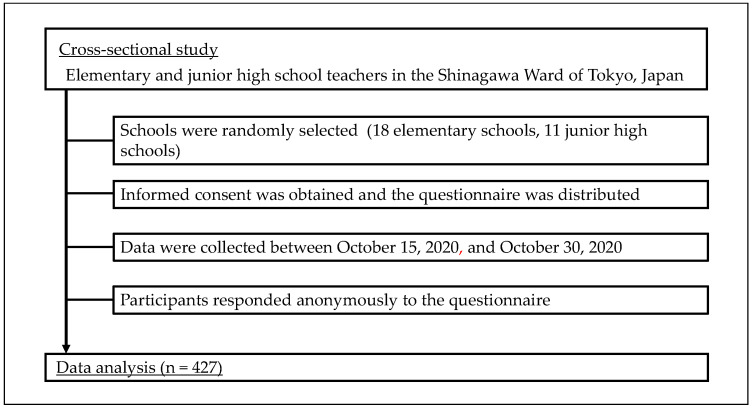
Flow-chart of the cross-sectional study.

**Table 1 ijerph-19-07568-t001:** Participants’ characteristics (n = 427).

Variables	
School Type (n = 427)	
Elementary school	305 (71.4 %)
Junior high school	122 (28.6 %)
Sex (n = 418)	
Male	157 (37.6 %)
Female	261 (62.4 %)
Age (n = 403)	
20–29 years	92 (22.8%)
30–39 years	139 (34.5%)
30–49 years	79 (19.6%)
50–59 years	64 (15.9%)
>60 years	29 (7.2%)

**Table 2 ijerph-19-07568-t002:** Changes in psychological conditions before and after school closures due to COVID-19.

	Elementary School		Junior High School		*p*-Value
	Male Teacher	Female Teacher	Male Teacher	Female Teacher	
Feeling high (n = 409)					0.82
Improved	11 (12.2%)	20 (9.9%)	7 (10.8%)	4 (7.7%)	
No Change	63 (70.0%)	143 (70.8%)	49 (75.4%)	35 (67.3%)	
Deteriorated	16 (17.8%)	39 (19.3%)	9 (13.8%)	13 (25.0%)	
Irritation (n = 410)					0.06
Improved	3 (3.4%)	24 (11.8%)	11 (16.9%)	4 (7.5%)	
No Change	66 (74.2%)	129 (63.5%)	40 (61.5%)	31 (58.5%)	
Deteriorated	20 (22.5%)	50 (24.6%)	14 (21.5%)	18 (34.0%)	
Anger (n = 408)					0.42
Improved	3 (3.4%)	22 (10.9%)	7 (10.8%)	4 (7.7%)	
No Change	73 (82.0%)	146 (72.3%)	47 (72.3%)	38 (73.1%)	
Deteriorated	13 (14.6%)	34 (16.8%)	11 (16.9%)	10 (19.2%)	
Indignation (n = 409)					0.10
Improved	3 (3.4%)	17 (8.4%)	5 (7.7%)	2 (3.8%)	
No Change	71 (79.8%)	155 (76.4%)	50 (76.9%)	33 (63.5%)	
Deteriorated	15 (16.9%)	31 (15.3%)	10 (15.4%)	17 (32.7%)	
Anxiety (n = 411)					0.73
Improved	9 (10.1%)	26 (12.8%)	6 (9.2%)	8 (14.8%)	
No Change	55 (61.8%)	106 (52.2%)	40 (61.5%)	29 (53.7%)	
Deteriorated	25 (28.1%)	71 (35.0%)	19 (29.2%)	17 (31.5%)	
Chagrin (n = 411)					0.24
Improved	7 (7.9%)	24 (11.8%)	10 (15.4%)	8 (14.8%)	
No Change	69 (77.5%)	135 (66.5%)	44 (67.7%)	31 (57.4%)	
Deteriorated	13 (14.6%)	44 (21.7%)	11 (16.9%)	15 (27.8%)	
Impotent feeling (n = 409)					0.33
Improved	6 (6.7%)	16 (8.0%)	8 (12.3%)	4 (7.4%)	
No Change	67 (75.3%)	149 (74.1%)	50 (76.9%)	35 (64.8%)	
Deteriorated	16 (18.0%)	36 (17.9%)	7 (10.8%)	15 (27.8%)	
Depression (n = 412)					0.499
Improved	4 (4.4%)	19 (9.4%)	8 (12.3%)	5 (9.3%)	
No Change	63 (70.0%)	129 (63.5%)	45 (69.2%)	34 (63.0%)	
Deteriorated	23 (25.6%)	55 (27.1%)	12 (18.5%)	15 (27.8%)	
NSOR (n = 411)					0.86
Improved	8 (8.8%)	26 (12.9%)	10 (15.4%)	5 (9.4%)	
No Change	72 (79.1%)	146 (72.3%)	47 (72.3%)	41 (77.4%)	
Deteriorated	11 (12.1%)	30 (14.9%)	8 (12.3%)	7 (13.2%)	
NSOT (n = 412)					0.89
Improved	5 (5.5%)	19 (9.4%)	7 (10.8%)	5 (9.3%)	
No Change	77 (84.6%)	161 (79.7%)	51 (78.5%)	42 (77.8%)	
Deteriorated	9 (9.9%)	22 (10.9%)	7 (10.8%)	7 (13.0%)	
EAP (n = 412)					0.40
Improved	3 (3.3%)	9 (4.4%)	6 (9.2%)	5 (9.4%)	
No Change	82 (90.1%)	173 (85.2%)	53 (81.5%)	42 (79.2%)	
Deteriorated	6 (6.6%)	21 (10.3%)	6 (9.2%)	6 (11.3%)	
ICCOW (n = 413)					0.02
Improved	0 (0%)	8 (3.9%)	6 (9.2%)	6 (11.1%)	
No Change	78 (85.7%)	169 (83.3%)	49 (75.4%)	39 (72.2%)	
Deteriorated	13 (14.3%)	26 (12.8%)	10 (15.4%)	9 (16.7%)	
NLWTEWO (n = 412)					0.50
Improved	3 (3.3%)	13 (6.4%)	5 (7.7%)	4 (7.5%)	
No Change	74 (82.2%)	153 (75.0%)	54 (83.1%)	41 (77.4%)	
Deteriorated	13 (14.4%)	38 (18.6%)	6 (9.2%)	8 (15.1%)	

*p*-values were derived from Fisher’s exact test analysis. Abbreviations: NSOR = No sense of reality; NSOT = No sense of time; EAP = Emotions are paralyzed; ICCOW = I cannot concentrate on work at all; NLWTEWO = No longer wanting to engage with others.

**Table 3 ijerph-19-07568-t003:** Changes in physical symptoms before and after school closures due to COVID-19.

	Elementary School		Junior High School		*p*-Value
	Male Teacher	Female Teacher	Male Teacher	Female Teacher	
Insomnia/nightmares (n = 413)					0.009
Improved	1 (1.1%)	16 (7.8%)	2 (3.2%)	5 (9.3%)	
No Change	82 (90.1%)	152 (74.1%)	56 (88.9%)	39 (72.2%)	
Deteriorated	8 (8.8%)	37 (18.0%)	5 (7.9%)	10 (18.5%)	
Palpitations (n = 413)					0.13
Improved	1 (1.1%)	12 (5.9%)	1 (1.6%)	0 (0%)	
No Change	86 (94.5%)	173 (84.4%)	59 (92.2%)	48 (90.6%)	
Deteriorated	4 (4.4%)	20 (9.8%)	4 (6.2%)	5 (9.4%)	
Standing dizziness (n = 414)					0.03
Improved	1 (1.1%)	16 (7.8%)	2 (3.1%)	0 (0%)	
No Change	85 (93.4%)	169 (82.4%)	57 (89.1%)	46 (85.2%)	
Deteriorated	5 (5.5%)	20 (9.8%)	5 (7.8%)	8 (14.8%)	
Sweating (n = 414)					0.83
Improved	1 (1.1%)	7 (3.4%)	1 (1.6%)	1 (1.9%)	
No Change	85 (93.4%)	182 (88.8%)	56 (87.5%)	49 (90.7%)	
Deteriorated	5 (5.5%)	16 (7.8%)	7 (10.9%)	4 (7.4%)	
Dyspnea (n = 414)					0.93
Improved	1 (1.1%)	5 (2.4%)	1 (1.6%)	0 (0%)	
No Change	87 (95.6%)	190 (92.7%)	59 (92.2%)	51 (94.4%)	
Deteriorated	3 (3.3%)	10 (4.9%)	4 (6.2%)	3 (5.6%)	
Digestive symptoms (n = 414)					0.55
Improved	1 (1.1%)	7 (3.4%)	0 (0%)	1 (1.9%)	
No Change	85 (93.4%)	176 (85.9%)	58 (90.6%)	48 (88.9%)	
Deteriorated	5 (5.5%)	22 (10.7%)	6 (9.4%)	5 (9.3%)	

*p*-Value derives from Fisher’s exact test analysis.

**Table 4 ijerph-19-07568-t004:** Impact of COVID-19 on work before and after school closures.

	Elementary School		Junior High School		*p*-Value
	Male Teacher	Female Teacher	Male Teacher	Female Teacher	
EIITW (n = 406)					0.37
Improved	6 (6.7%)	18 (9.0%)	4 (6.2%)	6 (11.5%)	
No Change	76 (85.4%)	154 (77.0%)	54 (83.1%)	36 (69.2%)	
Deteriorated	7 (7.9%)	28 (14.0%)	7 (10.8%)	10 (19.2%)	
LOTTA (n = 408)					0.80
Improved	7 (7.9%)	13 (6.5%)	5 (7.7%)	3 (5.6%)	
No Change	70 (78.7%)	154 (77.0%)	50 (76.9%)	38 (70.4%)	
Deteriorated	12 (13.5%)	33 (16.5%)	10 (15.4%)	13 (24.1%)	
LOTCA (n = 403)					0.79
Improved	6 (6.8%)	11 (5.6%)	3 (4.6%)	4 (7.4%)	
No Change	66 (75.0%)	151 (77.0%)	53 (81.5%)	37 (68.5%)	
Deteriorated	16 (18.2%)	34 (17.3%)	9 (13.8%)	13 (24.1%)	
LOTWE (n = 406)					0.31
Improved	1 (1.1%)	11 (5.5%)	3 (4.6%)	3 (5.7%)	
No Change	73 (83.0%)	150 (75.0%)	46 (70.8%)	36 (67.9%)	
Deteriorated	14 (15.9%)	39 (19.5%)	16 (24.6%)	14 (26.4%)	

Abbreviations: EIITW = Excessively immersed in the work; LOTTA = Lowering of the thinking ability; LOTCA = Lowering of the concentration ability; LOTWE = Lowering of the work efficiency. *p*-values were derived from Fisher’s exact test analysis.

**Table 5 ijerph-19-07568-t005:** Stress causes for teachers associated with COVID-19 infection control.

	Elementary School		Junior High School		*p*-Value
	Male Teacher	Female Teacher	Male Teacher	Female Teacher	
AWM (n = 413)					0.04
No	29 (31.9%)	50 (24.6%)	27 (41.5%)	12 (22.2%)	
Yes	62 (68.1%)	153 (75.4%)	38 (58.5%)	42 (77.8%)	
BTAHSG (n = 413)					0.54
No	67 (73.6%)	144 (70.9%)	52 (80.0%)	41 (75.9%)	
Yes	24 (26.4%)	59 (29.1%)	13 (20.0%)	13 (24.1%)	
DWBT (n = 412)					0.03
No	42 (46.2%)	91 (45.0%)	25 (38.5%)	13 (24.1%)	
Yes	49 (53.8%)	111 (55.0%)	40 (61.5%)	41 (75.9%)	
Social distance (n = 413)					0.71
No	59 (64.8%)	138 (68.0%)	43 (66.2%)	40 (74.1%)	
Yes	32 (35.2%)	65 (32.0%)	22 (33.8%)	14 (25.9%)	
ARDC (n = 413)					0.05
No	23 (25.3%)	74 (36.5%)	28 (43.1%)	24 (44.4%)	
Yes	68 (74.7%)	129 (63.5%)	37 (56.9%)	30 (55.6%)	
UFS (n = 413)					0.08
No	68 (74.7%)	143 (70.4%)	56 (86.2%)	40 (74.1%)	
Yes	23 (25.3%)	60 (29.6%)	9 (13.8%)	14 (25.9%)	
POROPAC (n = 413)					0.04
No	35 (38.5%)	99 (48.8%)	36 (55.4%)	33 (61.1%)	
Yes	56 (61.5%)	104 (51.2%)	29 (44.6%)	21 (38.9%)	
EWTNR (n = 413)					0.42
No	42 (46.2%)	113 (55.7%)	32 (49.2%)	26 (48.1%)	
Yes	49 (53.8%)	90 (44.3%)	33 (50.8%)	28 (51.9%)	
DNLCR (n = 413)					0.62
No	70 (76.9%)	153 (75.4%)	48 (73.8%)	45 (83.3%)	
Yes	21 (23.1%)	50 (24.6%)	17 (26.2%)	9 (16.7%)	

Abbreviations: AWM = Always wearing mask; BTAHSG = Body temperature and health status grasping; DWBT = Disinfection work by teachers; ARDC = Activity restriction during class; UFS = Using face shield; POROPAC = Prohibition or restriction of play and conversation; EWTNR = Eat without talking /no refills; DNLCR = Do not let children serve. *p*-values were derived from Fisher’s exact test analysis.

## Data Availability

The data is not publicly available as all participants have not consented to the public disclosure of the data online. However, the data presented in this study are available on request from the corresponding author.

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
