# Peer review of "Psychological and Physical Changes Caused by COVID-19 Pandemic in Elementary and Junior High School Teachers: A Cross-Sectional Study"

_ijerph, 2022, doi:10.3390/ijerph19137568_

Round 1
Reviewer 1 Report
The paper addresses a highly significant topic and the results of the research certainly provide an important addition to the scientific literature.
I feel that the "Conclusions" should start from line 304. The authors could further emphasise how the pandemic has been an occasion for research on issues that go beyond one specific cause.
I believe that the quality of the presentation could be further improved by some linguistic and textual revision revision, so that the results and the discussion emerge more clearly. This would involve rewriting some sentences ( examples include - line 69 should be "Teachers are required to provide support ..." or lines 295-298 the difference between "stressed" and "stressful") and also splitting some long paragraphs (example paragraph beginning on line 272).
The authors should also standardise the use of italics and quotation marks when referring to the items of questionnaire and the data collected.
The terms "gender" and "sex" are also mixed in the text. Personally I could suggest using only "sex", since the study is only concerned with biological sex and not with questions related to gender characteristics, roles or stereotypes (at least only marginally - reference to male teachers who "yell").
Author Response
Dear Reviewer1,
I attached reviwer comment.
Please see the attachment.
Sincerely
Nobuyuki Wakui (Corresponding Author)

Reviewer 2 Report
Authors used a cross-sectional study to identify difference of psychological and physiological changes due to the COVID-19 pandemic among elementary and junior high school teachers. Because the COVID-19 pandemic affects works of teachers as well as behavior of students, it may be meaningful to analyze the psychological and physiological change among teachers. However, this article has not been fully answered some of questions due to the insufficient description.
First, authors used only basic statistical techniques such as Fisher’s exact test, but it may be better for easy understanding by readers to use analytical statistics (e.g., logistic regression model with interaction term of gender and schools). Although the effect of the COVID-19 pandemic was divided into (1) effect of gender, (2) effect of school type (i.e., characteristics of schools), and (3) their interaction, it is difficult for readers to understand each effect only by the percentage of each category. Moreover, age of teachers may affect the psychological and physiological change, it better to use multivariable regression models for adjustment of potential confounding (e.g. age). Of course, as readers can calculate the odds ratios using the statistics in table 2-5, this manuscript may be useful as materials of future studies, but, if possible, authors should add results of logistic regression analysis. If authors do not add the result of logistic regression models, authors should add limitations in the discussion sections.
Second, it is difficult to understand some of description probably due to writing and English editing. For example, authors should add name of country (i.e., Japan) after the word “Tokyo” (P2L82), and it is difficult to understand “silent meals” (P2L48). Authors should edit the manuscript more carefully.
Minor comments.
P1L42. “three densities” may be “three Cs”.
P2L82. “Shinagawa area” may be “Shinagawa Ward”.
P2L86. “Over 15 days” may be “Over 16 days”.
P3L135. “school type and sex” may be “school type and gender”.
Table 2-5. “Men teacher” may be “Male teacher”.
Author Response
Dear Reviewer2,
I attached reviwer comment.
Please see the attachment.
Sincerely
Nobuyuki Wakui (Corresponding Author)

Reviewer 3 Report
This study aimed to determine psychological and physiological differences in elementary and junior high school teachers during COVID-19.
The authors proposed a study based on a questionnaire submitted to the community.
This questionnaire-based cross-sectional:
a) Was conducted among 427 teachers in Tokyo, Japan, (between October 15 and 30, 2020).
b) explored school type (elementary and middle schools), gender, age, and COVID-19 changes (psychological changes, physical changes, impact on work, and infection control issues perceived to be stressed).
The authors analyzed psychological changes, physiological changes, and work effects based on graded questions.
Regarding stress situation due to implementation of COVID-19 infection control, there were significant differences for disinfection work by teachers between male teachers in elementary school female teachers in junior high school and female teachers in elementary school female teachers in junior high school.
The authors concluded that the COVID-19 produced differences in psychological and physiological changes between male and female teachers in elementary and junior high schools.
The study is interesting and adds an important contribute to the scientific literature.
These are my minor comments for the authors, provided with a pure academic spirit.
1. The abstract must be rewritten better summarizing the sections of the manuscript, as for example the conclusions.
2. The aim is at the end of the introduction with the text “Therefore, this study conducted a cross-sectional survey of elementary and junior high school teachers to clarify the physiological and psychological differences during the COVID-19 pandemic”. I suggest to expand and clarify it better with more details. For example which phycological and physiological features …the authors are looking at?
3. Avoid in the methods paragraphs of 1 or 2 sentences.
4. Insert in the methods a flow-chart
5. Was the questionnaire submitted electronically? Please add this information and a print screen of the survey, perhaps in appendix.
6. Introduce the themes of the results with a few sentences and check the resolution and the fitting on the MDPI standards of the tables.
7. Minimize the acronyms and insert them in a table.
Author Response
Dear Reviewer3,
I attached reviewer comment.
Please see the attachment.
Sincerely,
Nobuyuki Wakui

Round 2
Reviewer 2 Report
Authors revised manuscript, but the manuscript still has description to be revised.
MINOR COMMENT.
P2L69.I could not understand the meaning of the words “school-aged”.
Author Response
Dear Reviewer 2
Thank you very much for reviewing our manuscript.
We attached revised manuscript.
Please see the attachment.
Sinserely,
Nobuyuki Wakui (Corresponding Author)
